# Assessment of Vapor Formation Rate and Phase Shift between Pressure Gradient and Liquid Velocity in Flat Mini Heat Pipes as a Function of Internal Structure

**DOI:** 10.3390/mi14071468

**Published:** 2023-07-21

**Authors:** Ioan Mihai, Cornel Suciu, Claudiu Marian Picus

**Affiliations:** Faculty of Mechanical Engineering, Automotive, and Robotics, Stefan cel Mare University, 720229 Suceava, Romania; claudiu.picus@usm.ro

**Keywords:** microchannels, mini flat MHP’s, pressure gradient, porous medium

## Abstract

Flat mini heat pipes (FMHPs) are often used in cooling systems for various power electronic components, as they rapidly dissipate high heat flux densities. The main objective of the present work is to experimentally investigate whether differences in the rate of vapor formation occur on an internal structure containing trapezoidal microchannels and porous sintered copper powder material. Several parameters, such as hydraulic diameter and fluid velocity through the material, as a function of the internal structure porosity, were determined by calculation for a steady state regime. Reynolds number was determined as a function of porosity, according to Darcy’s law, and the Nusselt number was calculated. Since the flow is Darcy-type through the porous medium inside the FMHP, the Darcy friction factor was calculated using five methods: Colebrook, Darcy–Weisbach, Swamee–Jain, Blasius, and Haaland. After experimental tests, it was found that when the porous and trapezoidal microchannel layers are wetted at the same time, the vaporization progresses at a faster rate in the porous material, and the duration of the process is shorter. This recommends the use of such an internal structure in FMHPs since the manufacturing technology is simpler, the materials are cheaper, and the heat flux transport capacity is higher.

## 1. Introduction

To date, a multitude of micro flat heat pipes has been developed with different internal structures for cooling various systems, including power electronics. So far, mainly three types of heat pipes have been made: grooved, sintered, or composited heat pipes (including FMHPs). A classification of heat pipes [1], first by construction principle and by the structure of the internal capillary layer, is shown in Figure 1.

In addition to the relatively low cost of FMHPs, the low dimensions, and high heat flux dissipation power, the literature [2] shows that FMHPs have various typical applications of microelectronic devices. According to [3], electronic devices work at temperatures usually between −5 ÷ 65 °C, but recently, the maximum heat flux in these components is increasing more and more, reaching maximum temperatures of 70 ÷ 80 °C. Increases above a certain threshold can cause functional instability or even destruction of electronic devices. To avoid this, one of the cooling methods uses thermal tubes. A great diversity of thermal tubes has appeared, as shown by [2,4,5,6], who mentions the use of FMHPs in electronics cooling, space thermal control, aircraft devices, traction drives, audio amplifiers, in cooling of closed cabinets in harsh environmental conditions, space applications under vacuum conditions [4,7] nanofluidic technologies in medicine, process engineering, avionics applications, solar photovoltaic/loop-heat-pipe, (PV/LHP) heat pump system [8] electronic applications, solar units, and aircraft [9] self-driving car, smart TV, smart grid, aerospace, radar, camera, computer, cellphone [10], solar energy systems, heat recovery systems, air conditioning systems, cooling of energy storage and electronic equipment, industrial applications and space apparatus.

Similarly, Ref. [11] states that MHPs are of interest for use in implanted neural pacemakers, sensors and pumps, electronic wristwatches, active transponders, self-powered temperature displays, and temperature warning systems. Recently, thermal tubes have also found applications in the fields of Energy, Global Warming & Environment, and Healthcare [12].

According to M V Pukhovoy et al. [13], with increasing heat dissipation power of power electronic components, which often reach heat flux densities of the order of $1000\;W/{\mathrm{cm}}^{2}$, the standard dimensions of microelectronic elements are reduced to 7 nm. Devices made of silicon carbide or gallium nitride have been developed, with chip element sizes reaching 6÷9 nm. In these specific cases, heat flux densities in a range of $100÷300\;W/{\mathrm{cm}}^{2}$ are achieved for surface areas as small as $4\;{\mathrm{mm}}^{2}$ and $10\;W/{\mathrm{cm}}^{2}$ for $1\;{\mathrm{cm}}^{2}$ [2,12,13,14,15]. In the case of FMHP [2], for copper-water coupling, heat fluxes of the order of 40 W/cm^2^ applied to the evaporator wall in a vertical position can be achieved. This paper combines analytical analysis for a number of functional parameters of FMHP with experimental studies of their behavior when using a porous wick structure with trapezoidal microchannels. The porosity of the material denoted by Φ, which in this case is part of a packed bed, according to [16], represents the ratio of the volume occupied by the pores to the total volume of the assembly. The choice of the inner capillary layer for the FMHP depends on several factors, but the most important is the compatibility of the material with the properties of the working fluid. It is well known that the role of the capillary layer is to generate an internal capillary pressure which, together with the adhesion forces, leads to the movement of the working fluid through the FMHP. The choice of the type of capillary layer is secondly aimed at its ability to evenly distribute the working fluid in the vaporization zone. Providing both functions requires the use of capillary layers with different shapes, especially for FMHPs where operation requires horizontal placement. In this case, the return of the condensed liquid to the evaporator is no longer produced gravitationally, the major role of the capillary layer being to distribute the working liquid circumferentially. Another characteristic of the capillary layer that needs to be optimized is its thickness. The heat transport capacity of an FMHP increases as the thickness of the capillary layer increases up to an optimum maximum value. However, the increase in the radial thermal resistance of the capillary layer due to the increase in thickness is contrary to the increase in transport capacity and leads to a reduction in the maximum heat flux from the evaporator. The total thermal resistance of the vaporizer also depends on the conductivity of the working fluid in the capillary layer. Another necessary property to be fulfilled by the capillary layer is its compatibility with the wettability of the working fluid, which must allow constructive adaptability to the shape of the inner wall of the FMHP. In the case of FMHP, there are several materials that can be used for inner capillary layers. In order to achieve the most optimal heat transport characteristics, a compromise between the thickness of the capillary layer size and the structure of the inner pores (sintered capillary layers) is necessary. In this way, the inhomogeneity of the porous capillary layer assimilated to the pore size will result in a total thermal conductivity for the whole assembly, which depends equally on the working fluid, the material from which it is made, and the number of pores per unit volume. Experiments carried out by Peter, H.J. de Bock et al. [17] have shown that the use of sintered capillary layers, in combination with micro-channels distributed longitudinally along the direction of vapor travel in certain operating situations, can reduce the drying of the vaporization zone. However, even in this case, situations may arise where the variation of the heat flux may be large enough to cause a blockage of the flow of accumulated condensate from the condenser to the vaporizer. In two-phase heat transfer, the heat transfer performance of an FMHP is strongly influenced by the working fluid fill ratio and the space left free for vapor circulation. If the free space through which the vapor flows during heat transfer is greatly reduced, the un-evaporated liquid remaining in the capillary layer tends to accumulate in the corners and at the edges of the FMHP, thus reducing the overall thermal resistance of the system. Under the same conditions, there is a reduction in the radius of curvature of the liquid meniscus in the condenser. This has been experimentally found by F. Lefèvre et al. [18], who showed that if the space through which the vapor flow decreases by 1 mm, the heat transfer capacity of the FMHP decreases by up to 35%. An important condition in the design of an FMHP is the capillary boundary. The threshold at the capillary limit is reached when vapor and liquid exceed the thickness of the capillary layer, the internal pressure prevents the movement of vapor and liquid relative to each other, and if the absorption capacity of a given porous material can no longer provide for fluid circulation within its pores. The maximum thickness of the sintered capillary layer is strictly related to the size of the voids in the layer or, more precisely, to the size of the microspheres making up the layer. In the case of FMHP, this capillary boundary phenomenon prevents the movement of freshly condensed liquid from the condenser to the vaporization zone. As in the other cases presented above, this also leads to the “drying out” of the vaporization zone during the heat transfer that takes place through the micro-tube.

## 2. Computational Details

### 2.1. Influence of Polysynthetic Medium Porosity on FMHP’s Surface Layer Heat Transfer

The porous material obtained and used in our own laboratory for FMHP is sintered copper powder. The simplest method to study heat transfer through porous media is that of heat transfer under the assumption that the fluid diffuses into the porous medium and remains at rest, being traversed by a heat flux ϕ [19,20]. If, by initial boundary conditions, the temperature of the fluid (index “l”) and that of the solid (index “s”) in the transient regime is considered, the heat transfer equation for the two media will be:(1)$${\left(\rho c\right)}_{l\_s}\frac{\partial T}{\partial t}=\nabla \left(h_{l\_s}\nabla T\right).$$

In Equation (1), the following notations are made, ρ—density of the working fluid, *c*—specific heat, *T*—temperature, *t*—time, ∇—Nabla differential operator, *h*—heat conduction coefficient. At the fluid (index “fl”)—porous material (index “ps”) interface, the ${\left(\rho c\right)}_{l\_s}$ term can be defined as follows:(2)$${\left(\rho c\right)}_{l\_s}=\phi \rho_{fl}c_{fl}+\left(1-\phi \right)\rho_{ps}c_{ps},$$
and represents an average value of the specific heat for the two environments.

This approach can only be applied for the condition when the transient thermal regime is short and when the ratio between the thermal conductivity of the liquid and the solid is much different from unity. If the solid and the liquid are not in thermal equilibrium, a system consisting of two equations is considered. For ease of calculation, a rigid medium with a fixed structure is considered. The liquid in the porous medium is assumed to have constant viscosity, density, and thermal conductivity. Since the liquid is considered incompressible, by applying an external temperature, the force it exerts on the porous medium can be characterized by the energy equation [21]:(3)$$\rho_{l}c_{l}\frac{\partial T}{\partial t}+\rho_{l}c_{l}\nabla \left(\vec{u}T\right)=h_{l}\nabla^{2}T,$$
where $\vec{u}$ is the velocity vector in the direction of fluid flow. Of course, in this case, heat transfer cannot be minimized by the viscosity of the interacting fluid media. This can only be considered when the flow is predominantly in one direction (unidirectional flow) or for boundary layer flow.

In the analysis of convective heat transfer through porous media, it should be taken into account that the most important transformations occur at the interface of these media with the wall of the FMHP. The general equation of convective heat transfer [21] from the solid medium (FMHP wall) to the working fluid diffusing into the porous medium can be written as:(4)$$q_{s\_l}=h_{s\_l}a_{s\_l}\left(\bar{T_{s}}-\bar{T_{l}}\right).$$

In Equation (4), the area of the solid-liquid interface has been denoted by $a_{s\_l}$ and was determined to be $a_{s\_l}=0.015{\;m}^{2}$, while $h_{s\_l}$ represents the convective solid-liquid heat transfer coefficient. An analysis of heat transfer in porous media shows that the two forms of heat transfer, convective and conductive, cannot be separated as the interface between the two media determines this. These two effects, convective and conductive, cause a volumetric force to arise, determined by the porosity coefficient of the medium, the hydraulic diameter of the FMHP, and the macroscopic parameters of the media.

The hydraulic diameter $D_{h}$, can be determined by the relation [22]:(5)$$D_{h}=\frac{4\;D_{m}\;\Phi_{p}}{C_{s}\left(1-\Phi_{p}\right)}.$$

In Equation (5), the hydraulic diameter of the microfluid flow channels was noted as $D_{h}=4R_{h}$, where $R_{h}$ is the hydraulic radius, $D_{m}=0.456\times 10^{-3}m$ is the grain diameter, porosity was considered $\Phi_{p}=0.25÷0.75$, and $C_{s}$ is the shape constant of the cross-section, which is determined as [22]:(6)$$C_{s}=\frac{m_{fl}\mu_{fl}}{\gamma_{W}R_{h}{}^{2\;i\;a_{po}}}.$$

In Equation (6), the fluid flow rate is denoted by $m_{fl}$ and the fluid viscosity by $\mu_{fl}$.

The calculations were performed using a code developed under Mathcad14.

In order to determine the flow regime estimated with the Reynolds number, it is necessary to determine the flow velocity of the working fluid through the porous material [23].
(7)$$w_{l}=\frac{\Phi_{p}}{\mu_{fl}}\left(\frac{D_{h}^{2}}{16C_{k}}\right)\left(\frac{\Delta P}{L_{hp}}\right){\left(\frac{L_{hp}}{L_{e}}\right)}^{2}.$$

The notations in the fluid velocity relation [23] have the following meanings: $\mu_{fl}$—fluid viscosity, $D_{h}$—hydraulic diameter, $C_{k}$—Kozeny constant, ΔP—pressure difference, $L_{hp}$—medium length, $L_{e}$—actual flow length. In microthermal tubes, it is also possible to consider the phenomenon of convective diffusion, which occurs due to free convection and occurs as a result of density differences induced by changes in pressure and inner temperature. Diffusion will be analyzed by determining the velocity of fluid movement through porous media. The evolution of the hydraulic diameter and the fluid flow velocity through the porous medium in dependence on the porosity of the polysynthetic medium is shown in Figure 2.

The increase in porosity obviously leads to an increase in the hydraulic diameter (case a) and the flow velocity of the working fluid (b), which can be seen in Figure 2.

Knowing the average value for grain diameter $D_{g}=1.78\times 10^{-3}m$, obtained by measurements, one can calculate the dynamic specific surface area [22] with the relation:(8)$$a_{vd}=\frac{C_{s}}{D_{g}}$$

The Reynolds number will be determined using Equation (9) [22].
(9)Reh=ρfl wl4Φpavd(1−Φp)μfl.

Using the developed code, the Reynolds number variation as a function of the porosity of the polysynthetic material was determined, and the values obtained by calculation are shown in Figure 3.

The heat exchange process takes place through two different mechanisms. The Nusselt number is determined for the case of fluid diffused in the porous copper medium and for fluid flowing through polysynthetic material. The total heat transfer coefficient of the fluid flowing through the porous medium formed by copper microspheres expressed by interfacial Nusselt number $Nu_{s\_l}$ [24], can be given as a function of Prandtl and Reynolds numbers, as:(10)Nus_l=|Nus_l∗(1+aPrRehb+Nus_l∗Pr1−mReh1−n)   if   m=0.5   and   n>0.5hs_ldpkm   otherwise,
where Pr is the Prandtl number and Reh is the Reynolds number based on Darcy velocity and hydraulic diameter. Stagnant Nusselt number, $Nu_{s\_l}^{*}$, can be written as a function of Darcy’s laws as: $Nu_{sol,lic}^{*}=\frac{\sigma_{s\_l}}{\alpha_{hA}\sigma_{s|\_l}+\alpha_{hB}}$, where $\alpha_{hA}$ and $\alpha_{hB}$ represent the dimensionless conduction layer thickness in fluid and solid phases as normalized by the particle diameter, respectively σ, the solid to fluid thermal conductivity ratio. It was adopted $\sigma_{s\_l}=394\;W/\mathrm{mK}$, this value falling between the recommended values of $1<\sigma <10^{3}$. In Equation (10), a and b represent coefficients whose ratio is a constant, m=0.5, if Pr≪1 or m=0.33 if Pr≫1 and n>0.5 for large Reynolds numbers.

In the graph shown in Figure 4, the correlation between the Nusselt number and porosity was represented when the Prandtl number changes for convective transfer between the two capillary media. It should be noted that Petukhov [25] indicates values of the Prandtl number for a fully developed turbulent pipe flow ranging from 0.5 to 2000 while the Nusselt equation deduced by Gnielinski [22,26,27,28] for turbulent as well as transition regions for a smooth tube is valid only if 0.6<Pr<10.

The amount of liquid carried through the two media (copper microsphere surface and trapezoidal microchannels) is not the same. After the entire amount of liquid has evaporated from the microchannels, the liquid in the polysynthetic capillary layer continues to evaporate. The evaporation of the fluid is caused by conductive heating of the polysynthetic porous layer by the heat from the walls of the flat thermal micro-tube. The fluid moves through the capillary layer and starts to wet the inner porous cavities of the porous medium, creating a convective-type phenomenon. In this case, for porous media, the Nusselt number can be determined as a function of Reynolds number and/or Prandtl number by empirical relations proposed by various authors in different forms [13,26,27,28,29,30,31,32,33,34,35,36], at which the flow regime is turbulent.
(11)NuGn=(0.79ln(Reh)−1.64)−28(Reh−1000)Pr1+12.7((0.79ln(Reh)−1.64)−28)12(Pr23−1)

The Reynolds number determined by the mathematical model described above was used for the calculations. Gnielinski [37] notes the Darcy friction [36] f=(0.79ln(Reh)−1.64)−2. The variation of the Nusselt number is important for the study of heat transfer by the mechanisms described above. Starting from this consideration, a fluid Prandtl number of 0.5 to 2000 was used in calculations in the relation proposed by Gnielinski $Nu_{Gn\_A}$ [25], which provides a very accurate correlation (index A = accurate) for turbulent flow in porous media.
(12)NuGn_A=(0.79ln(Reh)−1.64)−28(Reh−1000)Pr1+12.7((0.79ln(Reh)−1.64)−28)12(Pr23−1)[1+(DhLHP)23](PrPrw)0.11

The Prandtl number at the wall was denoted by ${\mathrm{Pr}}_{w}$, knowing that $0.6<\mathrm{Pr}<10^{5}$ for Reynolds values 2300<Re<106. The results of the calculations performed for the Nusselt number with the Mathcad code using the two criterial relations are shown in Figure 5.

Nusselt calculated with the normal method as specified in the article is obtained with a calculation code that uses Equation (11), and the improved one uses Equation (12). The data obtained by calculation show different values of the Nusselt number in the area of the average porosity, but the curves are the same, and the heat exchange is favored if the porosity increases. It can be appreciated that in a micro heat pipe with a composite structure, in the capillary layer made of copper microspheres, the liquid vaporizes much faster than the liquid embedded in the porous material. In composite structures, however, there is an advantage (which will also be seen in the experimental determinations) in that in the porous capillary layer made of sintered copper powder, the vaporization process starts faster and continues after the completion of this process in the trapezoidal microchannels.

Thanks to the composite structure of FMHP, an additional amount of vapor is produced by the described mechanism. Even though the structure of the FMHP is more complex, the described phenomenon prevents drying of the vaporization zone in case of high heat fluxes and, at the same time, enhances heat transfer.

### 2.2. Determination of the Phase Difference between the Pressure Gradient and Liquid Velocity in the Boundary Layer of an FMHP

The analyzed thermal micro tube has in its structure two porous media, one made of sintered copper powder (obtained in the laboratory) and the other of trapezoidal microchannels. By displacing the liquid in the two porous media, transient inertial forces arise between the liquid and the porous structure. The transient inertia force will be all the more important as there will be a phase difference between the oscillation velocity and the pressure gradient. If the velocity amplitude for the fluid displacement through the boundary layer is taken into account [24], the maximum pressure gradient coefficient $\hat{f}$ can be written as:(13)f^=|φStReh+φBReh122.64π+2.67φIπ+φMReh12idhidAfld+(1+φV)idhidAfld|.

In Equation (13), $\varphi_{St}$ represents the Stokes frictional force, $\varphi_{B}$ is the frictional force due to the advection boundary layer, $\varphi_{I}$ is the inviscid form drag, $\varphi_{M}$ represents the Basset memory viscous force due to the transient boundary layer, $\varphi_{V}$ is the inviscid virtual mass force, and $A_{fld}$ denotes the amplitude of the fluid displacement of the superficial flow which has the expression $A_{fld}=\Phi \bar{A}$ where $\bar{A}$ represents the intrinsic average of the fluid displacement in the pores. The offset between pressure and velocity is determined as:(14)θp_w=1tan((1+φV)dhAfl+φM2RehdhAflφSReh+φBReh2.64π+2.67φIπ+φM2RehdhAfl)

The relation given in Equation (14) shows that the pressure gradient of an oscillating flow in a porous medium depends on the Reynolds number as well as on the ratio of the hydraulic diameter $d_{h}$ $A_{fld}$. In order to represent the phase shift between the pressure gradient and the fluid velocity on the boundary layer, the values $\varphi_{St}=109\;N$, $\varphi_{B}=5\;N$, $\varphi_{V}=1.7\;N$, $\varphi_{I}=1\;N$ and $\varphi_{M}=25\;N$, were considered. The variation of the pressure gradient and the phase shift between the vapor pressure and the vapor displacement velocity depending on the porosity of the polysynthetic material obtained in Mathcad can be seen in Figure 6a,b.

As the evaporation of the liquid develops more and more and, the Reynolds number increases, and the pressure-velocity phase shift decreases, the amplitude of the boundary layer oscillations is reduced. In fact, the evaporation process in porous media is mainly due to convective heat transfer. Oscillations occurring in the boundary layer at the separation between two media made of different materials are caused by the nucleation phenomenon occurring in the liquid during the heat transfer process. The nucleation development will be visualized in the experimental part of this paper obtained by filming the vaporization zone for both the porous medium and the trapezoidal microchannels. One of the important factors in explaining the phenomenon is determined by the memory of the Basset viscous forces due to boundary layer transit. Vapor bubbles, which are produced inside the heated liquid, start to rise toward the boundary layer. The variable speed of the vapor bubbles, as they move through the liquid inside the FMHP, causes pressure on the boundary layer explained by the Basset viscous forces, which leads to the oscillation phenomenon, a process that diminishes as the liquid evaporates completely. At the moment of phase change, nucleation disappears, and thermodynamic equilibrium is reached. The study of oscillations due to the pressure gradient for the liquid boundary layer between the two porous media is necessary for the study of the vaporization occurring in the two layers. The different structure of the two layers leads to different behavior during heat transfer. By vaporizing the liquid in the first capillary layer, vapor bubbles of a certain temperature pass through the boundary layer, causing the second capillary layer to heat up. This results in a movement of the vapor bubbles from and towards the two layers, causing oscillations in the boundary layer, which are attributed to the existence of a pressure gradient.

### 2.3. Darcy Flow through a Porous Medium Inside a Horizontal Micro Heat Pipe

For the study of flow and heat transfer phenomena, it is of interest to determine the friction factor of the liquid flowing through the two porous media. This can be investigated using Colebrook’s equation. Solving this equation is quite difficult [38,39,40,41] as it requires the use of iterations. Colebrook’s equation allows the determination of the Darcy friction factor (also known as Darcy–Weisbach), which is necessary to calculate the pressure drop or pressure loss due to wall roughness. Over time several equations and methods of solving them have been developed. The Colebrook–White equation has the form:(15)1f=−2log(ϕ3.7Dh+2.51Rehf)

Different methods are used to solve this equation [38,39,40,41], among which we mention the use of a dedicated calculator. The method solves the Colebrook Equation by Hand in Excel, using a root-finding VBA subroutine, using the Moody diagram, by approximation, or by the direct method. For relation (15), the programming code proposed by [39] was adapted to the values of the thermal microtube with a porous structure. For a comparative study of the Darcy friction factor, a code built in Mathcad14 for Equation (16), presented by Jukka Kiijärvi [41], was used.
(16)fDW=2Δp DLhpρflVfl2;   fSJ=0.25[log(e/D3.7+5.74Reh0.9)]−2;  fBl  =   0.316Reh0.25;     fHa=1−1.8  log  [(e/D3.7)1.11+6.9Reh].

The meanings of the notations in Equation (16) are: f—Darcy friction factor, DW—Darcy–Weisbach, Δp—pressure loss, D—inner diameter of the heat pipe, $L_{hp}$—length of the heat pipe or part of the pipe, $\rho_{fl}$—density of the fluid, $V_{fl}$—flow velocity of the fluid, SJ—Swamee–Jain, e—the roughness of pipe, Bl—Blasius, Ha—Haaland.

Figure 7 plots the variation of friction coefficients for porous media made of copper and polysynthetic material as a function of material porosity (Figure 7a) and Reynolds number (Figure 7b).

The results shown in Figure 7 were obtained using two computational codes, one made by the authors in Mathcad and one proposed by Praks, P.; Brkić, D in Matlab [38]. As it can be easily seen from Figure 7, the curves have the same trend for the case of Haaland, Swamee Jain, Blasius, and Colebrook’s equations, but in terms of value, there are differences of a few percent. On the other hand, the model proposed by Darcy–Weisbach, in our opinion, gives values outside the tolerance field since the shape deviations are high and significant differences in value appear. In the paper, we have predicted the relationship between heat transfer and the friction factor of fluid flow through a porous medium. The results show that the rate of heat transfer through a porous structure in FMHPs increases with increasing Reynolds number and that there is a direct link between the Nusselt number and friction factor. The Darcy number decreased with increasing Reynolds number. This paper shows how the rate of vapor generation changes with changes in the internal structure of FMHPs.

## 3. Experimental Set-Up and Methodology

For experimental determinations, a set-up shown in Figure 8 was used, containing the following elements: 1—electronic module for controlling and monitoring FMHP parameters, 2—keypad for prescribing the voltage and current consumed by the heating resistors, 3—general on/off button, 4—vaporization zone heater voltage and current display, 5—cooler on/off button, 6—condensing zone cooling control system, 7—display for displaying the temperature read by thermocouples, 8—on/off switch thermocouple module, 9—data logger and connection with PC, 10—PC, 11—thermocouples, 12—heating block, 13—micro-thermal tube, 14—cooler, 15—electronic heater control module.

To study and visualize the behavior of the working fluid during the supply of liquid from the condensate, the initiation of vaporization, the development of the vaporization itself, and the displacement of the vapor, microscopic filming was carried out. The FMHP structure in the vaporization zone is visible in Figure 9 (made in the laboratory) and is composed of a porous medium made of sintered copper metal powder and trapezoidal copper microchannels. It is important to know that the filling of the inner structure with distilled water was performed at the same time. Over the porous layer and over the trapezoidal microchannels of the FMHP, a quartz glass was tightly applied to the top, then the tube was partially evacuated. A heating source consisting of electrical resistors fed from an adjustable power supply was placed under the micro heat pipe. Knowing the power parameters of the electrical resistors, the heat flux density can be determined by calculation.

A series of laboratory tests were carried out to verify the behavior of a micro heat pipe with a porous layer of sintered copper powder and trapezoidal microchannels. From the studies and observations, we found that the permeability of the porous capillary layer increases with increasing pore size. However, for homogeneous capillary layers, there is an optimal pore size that is relatively simple to determine. A trade-off is created between permeability and pore size. Porous capillary layers used in FMHP can have low performance if the pore sizes are between 80÷150 μm for horizontal operation where the working fluid displacement should not overcome the gravitational force and high-performance capillary layers where the pore sizes are between 30÷80 μm for vertical or inclined operation where pumping of the working fluid against gravity must be ensured. To transport the condensed working fluid from the condenser to the vaporizer, a capillary layer of a porous material made in the laboratory by sintering copper powder has been incorporated into the internal structure of the FMHP. As it is well known, if the liquid has to rise against gravity, it is recommended to use a capillary layer with low porosity. The maximum capillary pressure inside an FMHP (which is initially partially evacuated) is determined by the average pore size of the sintered capillary layer, decreasing as it increases. Sintered structures used in capillary layers provide higher capillary pressure than grooved or trapezoidal microchannel structures. However, the permeability of sintered capillary layers is lower. Therefore, there will be pressure losses due to friction. Both micro-channels and sintered layers must be made of the same material as the FMHP wall and be fixed to the inner walls of the FMHP in order to achieve the best thermal contact. In the case of the porous material, we have concluded from experimental determinations that it should be chosen with a relatively high degree of porosity in order to retain as much liquid as possible inside the pores. The absorption of liquid into the larger pores of the material ensures an increase in the capillary radius. The choice of material should be made in relation to the temperature at which the FMHP is to be used so that once the vaporization temperature of the working fluids is reached, the material does not change its physical properties. Also, the change in physical properties must not lead to the interaction of the material with the working fluids used.

## 4. Experimental Results and Discussion

The dimensions of the trapezoidal microchannels were obtained by laser profilometry using a NanoFocus μScan laser profilometer, a sample measurement of a microchannel is given in Figure 9a, and the details of the internal structure of the FMHP used in the experiments was obtained by the same method and can be seen in Figure 9b.

In the following paragraphs, reference is made to the polysynthetic material used in practice in the FMHP structure since its absorption capacity, internal structure, and compression deformation have superior quality parameters to other materials. Preliminary tests proved that the porous material retained its absorbent qualities when using distilled water, acetone as well as methanol. Figure 10a,b shows a microscopic image of two cases, namely the structure of the sintered material made of dry sintered copper powder (case a) and wetted with distilled water (case b).

In order to know the quantities used in calculations, laser profilometry was used to measure the surface of the porous material, both in a dry and fully wetted state. Figure 11 shows the results first scanned for the dry porous material.

In order to observe how the pore size changes when the material is wetted with the working fluid, it was necessary to scan it by laser profilometry, results illustrated in Figure 12 providing a series of physical parameters specific to this case.

For each case, the equipment creates a report from which an extract of the most important parameters for the two cases is shown in Table 1.

The experimental data obtained clearly show a change in the mean values of the microchannel parameters when using a porous material wetted with condensate due to the influence of the liquid weight on the pores. The values for pore area, diameter or size of microspheres or copper microparticles permeability, respective pore thread diameter of the porous material, and porosity were assumed, as shown in Table 2.

The experiments allowed to film at the same time two categories of internal structures (wick structure) of the FMHP. The results obtained can be seen in Figure 13, which includes a sample of the filmed frames.

In all the selected frames, it can be seen how in the first heating step and reaching the vaporization temperature, the vapor formed in the porous zone migrates to the trapezoidal microchannel zone. The velocity of fluid or vapor movement can be determined by calculations using the distance of fluid or vapor movement or development obtained on the basis of the micrometric scale of the images and the time on the basis of a specialized image processing editor (determined for the beginning and end of the processes).

Figure 14 shows extracts of detail frames for the porous zone obtained from the microscopy images.

The porous layer in the evaporation zone of frame 1350 in Figure 12 shows is initially dry, in which case, after liquid supply, the liquid pressure inside will be equal to the vapor pressure. As liquid displacement through the capillary layer (visible in frame 1360 right) takes place, the pressure inside the FMHP decreases, a phenomenon mainly due to friction of the working fluid with the internal structure of the capillary, gravity, and depending on the mass of liquid displaced. This reduction in capillary pressure is sufficient to cause the fluid to start boiling (nucleation) once condensate starts to reach the zone (frames 1370 to 1390). The pressure in the capillary layer is important because it causes nucleation to begin in the pore layer and at the wall of the FMHP. As this pressure decreases, the risk of boiling the liquid increases. There are several factors that can substantially influence the pressure in the capillary layer. Temperature is the determining factor as it affects the physical properties of the working fluid in the FMHP and is, in turn, controlled by the applied heat flux. In turn, the degree of permeability of the capillary layer influences the total pressure drop. Since the pressure loss in the capillary layer decreases with the length of the FMHP, the most likely place where fluid boiling can occur is at the top of the vaporization zone, which can be visualized in frames 1380 and 1390. If liquid boiling is avoided in this section, then this phenomenon will not occur along the entire length of the capillary layer. The occurrence of a pressure loss gradient is mainly attributed to the porosity of the capillary layer. For lower porosities, it has been found that the pressure increases faster than when the porosity tends towards maximum values. At porosities tending towards maximum values, more liquid accumulates in the porous material, which leads to a certain incompressibility of the porous material, in which case the capillary pressure loss is lower.

If the liquid vaporizes, the pressure in the vaporizer drops very much, but after leaving the vaporizer, the pressure in the adiabatic zone does not drop very much, and then the pressure in the condensation zone increases slightly due to the temperature reduction. An important influence on nucleation and boiling is the Reynolds number, which, when increasing, the capillarity of the polysynthetic starter decreases due to the increase in pressure by changing the vapor velocity, which causes the axial hydrostatic pressure to have an increasing tendency. An important condition for high capillarity is that the capillary layer has a high degree of permeability. This can be determined by its internal structure through the size of the pore size. A structure with the smallest pore size has a high degree of permeability, and as the porosity decreases, the capillary pressure decreases. The vaporization process continues for a long period (1400 to 1435 frames) with the development of new nucleation centers. After the nucleation phenomenon has fully developed, a flow of vapor formed through microchannels begins (frames 1440 to 1465) and then continues until a new wave of cold liquid condensate appears.

In order to study the vaporization phenomenon in a porous environment in comparison with that in a trapezoidal microchannel environment, movies were taken (an extract of frames is the one in Figure 15) on the same FMHP. The study allows a comparative analysis of vaporization and differentiation of the nucleation and vapor development mode for the two cases.

If, in the first case, corresponding to the porous layer, the vaporization started at frame 1360 and ended at frame 1465, in the case of trapezoidal channels, there is a phase shift compared to the case of micro trapezoidal channels. Although the liquid in the condensate covered the trapezoidal microchannels and the porous material at the same time, there is a delay from frame 1360 (porous layer) to 1435 (trapezoidal microchannel layer) where it is observed for the second case how nucleation starts and how the vaporization process begins to propagate.

## 5. Conclusions

In the present paper, various parameters, such as the hydraulic diameter and fluid velocity through the material as a function of the porosity of the internal structure, were determined by calculation for a steady state regime. Reynolds number was determined as a function of porosity, according to Darcy’s law, and the Nusselt number was calculated. Both are of interest in the study of heat exchange in an FMHP. The calculation of the Nusselt number was performed using the normal (using Equation (11)) and improved method (using Equation (12)) as a function of porosity and Prandtl number values. The determination of the phase difference between the pressure gradient and the liquid velocity in the boundary layer of a FMHP allows the analysis of how flow through the porous layer takes place. Since the flow is Darcy-type through the sintered copper powder porous medium inside the FMHP, the Darcy friction factor was calculated using five methods: Colebrook, Darcy–Weisbach, Swamee–Jain, Blasius, and Haaland. The results obtained were analyzed as a function of the change in porosity and Reynolds number.

The experiments were carried out on a surface structure composed of trapezoidal microchannels with an average base of 176.5 μm and a peak of 283.4 μm (see Figure 9, Figure 13 and Figure 15) and porous sintered copper powder material (see Figure 10, Figure 11 and Figure 12) with an average depth of 210 μm which is relevant for a comparative study as they have close average values. The internal structure of the porous material has a porosity ranging from 55% to 80% at a thickness that, in turn, ranges from 0.8 mm to 0.38 mm.

The heating system used provides a heat flux on the evaporator side of up to 140 W, which, used at maximum values, indicated that the vaporization process is in strict dependence on the internal structure of the FMHP. Theoretical calculations showed that when the maximum value of the porosity of the FMHP material is reached, the hydraulic diameter increases to 4.63 × 10^−3^ m, the Reynolds number to 4.585 × 10^5,^ and the Nusselt number to 1.67 × 10^3^, which favors the vaporization process by intensifying the convective phenomenon (if the Nusselt number increases, then the convective heat coefficient has the same tendency). The calculations showed that there is a phase difference between the pressure gradient and liquid velocity in the boundary layer of an FMHP.

The dimensional data used as input in the conducted calculations was obtained by measuring the surface microtopography by laser profilometry.

Several experimental tests were conducted by filming under the microscope the fluid feeding process of the internal structure of the FMHP. The condensed liquid was introduced into the FMHP, which was heated in the vaporization zone, followed by a nucleation process followed by vapor development. Knowing the time between frames and the space traveled by the vapor, it is possible to determine the vapor formation rate for both the porous medium and the trapezoidal microchannels. Using the same method, it was possible to estimate the velocity phase shift for the two interior structures, the porous one and the one with trapezoidal microchannels.

It was experimentally shown that for the porous medium structure versus trapezoidal microchannels, the vaporization proceeds faster from 5.44 ms to 5.86 ms versus a delayed onset at 5.74 ms to the end of vaporization at 5.84 ms. This shows that the vaporization process proceeds more efficiently in the porous sintered copper powder material layers subjected to this study. The explanation lies in the fact that the porous material performs better at high temperatures as it has a lower thermal resistance compared to the case of fluid flow through trapezoidal microchannels. This is due to the smaller thickness of the fluid film in relation to the contact surface, which in this case is larger in the porous material than in the case of trapezoidal channels. The study carried out experimentally proved that under certain operating conditions, it is possible that a porous material in an FMHP can transport heat more efficiently than a trapezoidal microchannel system.

In the present paper, an analysis was made on the heat transfer phenomena inside the FMHP, their limits, as well as the phase shifts that occur in the vaporization process depending on their inner structure.

## Figures and Tables

**Figure 1 micromachines-14-01468-f001:**
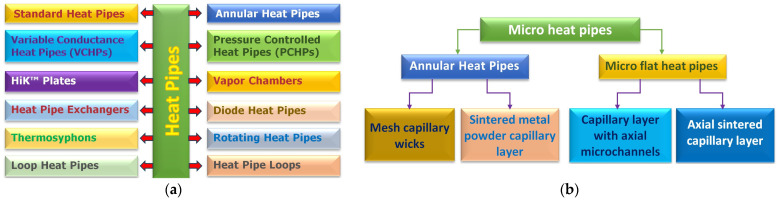
Heat pipes classification: (**a**) by construction principle and (**b**) by the structure of the internal capillary layer.

**Figure 2 micromachines-14-01468-f002:**
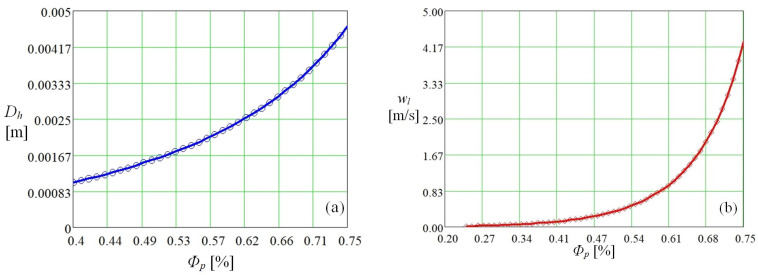
Variation of hydraulic diameter (**a**) and liquid velocity through a porous material (**b**) as a function of porosity.

**Figure 3 micromachines-14-01468-f003:**
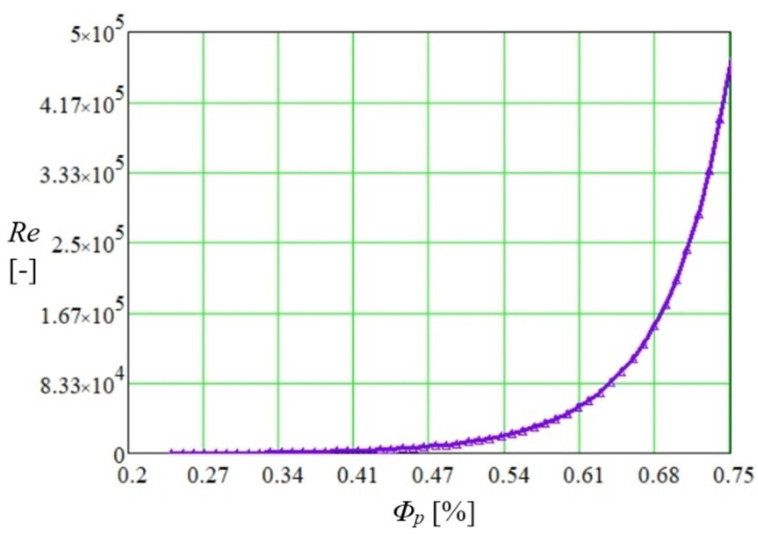
Variation of Re number with porosity of porous material in an FMHP.

**Figure 4 micromachines-14-01468-f004:**
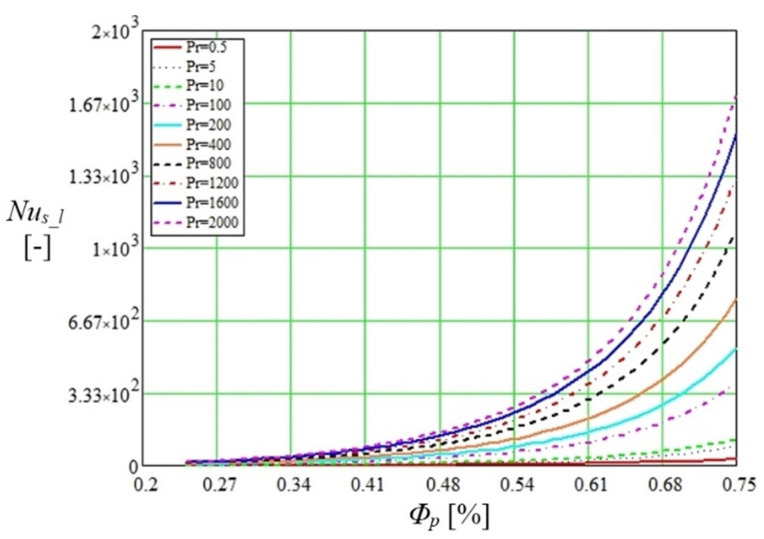
Nusselt representation as a function of porosity and Prandtl number values.

**Figure 5 micromachines-14-01468-f005:**
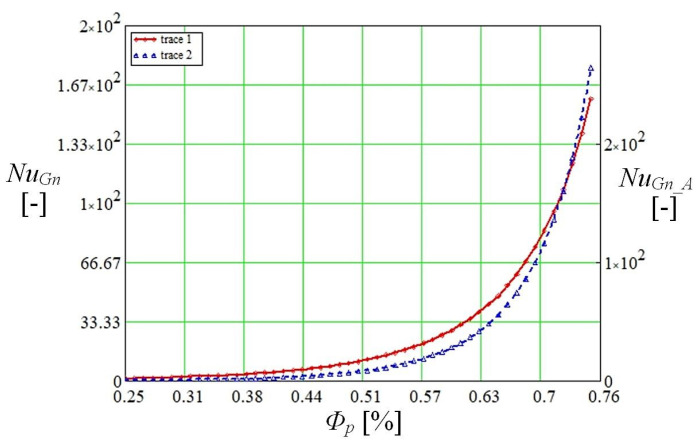
Variation of Nusselt number by normal (using Equation (11)) and accurate (using Equation (12)) method with porosity of porous layer.

**Figure 6 micromachines-14-01468-f006:**
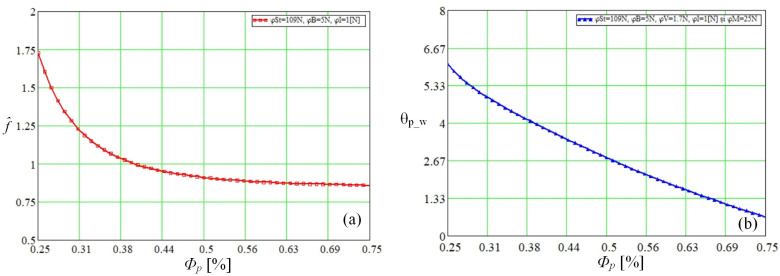
Maximum pressure gradient coefficient (**a**) and pressure-velocity gradient on the boundary layer (**b**) as a function of porosity.

**Figure 7 micromachines-14-01468-f007:**
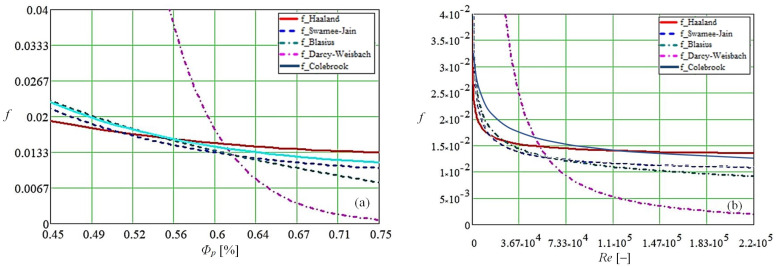
Friction coefficient through porous media as a function of porosity (**a**) and Reynolds number (**b**).

**Figure 8 micromachines-14-01468-f008:**
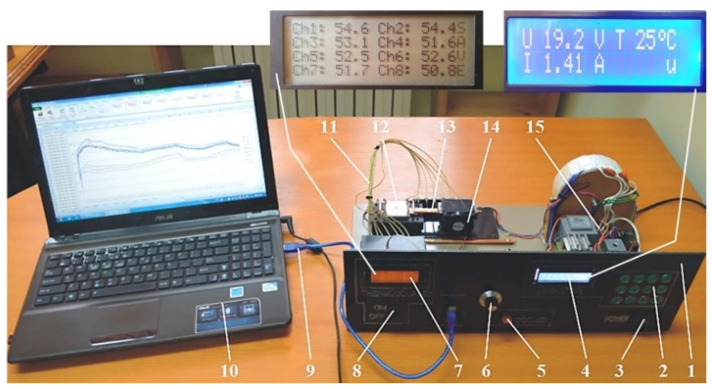
Experimental set-up for the study of FMHPs.

**Figure 9 micromachines-14-01468-f009:**
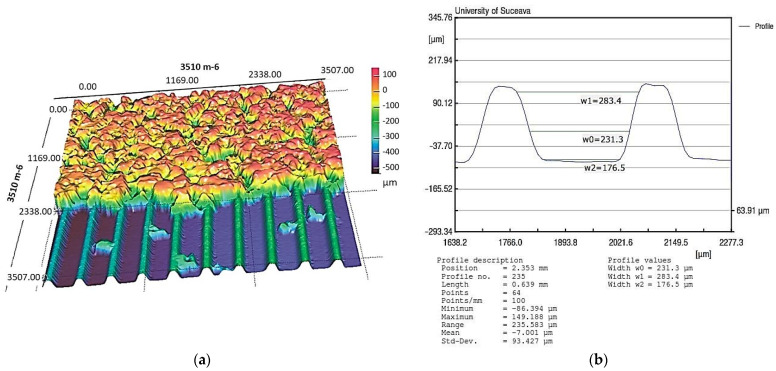
Sample measurement of trapezoidal microchannels: (**a**) 3D surface microtopography; (**b**) dimensional details for the microchannel transverse cross-section.

**Figure 10 micromachines-14-01468-f010:**
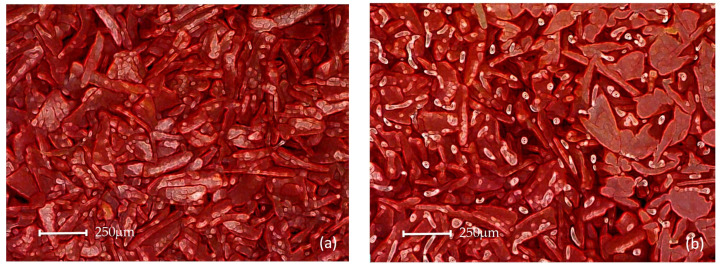
Porous sintered copper powder material: dry (**a**) and wetted with condensation liquid (**b**).

**Figure 11 micromachines-14-01468-f011:**
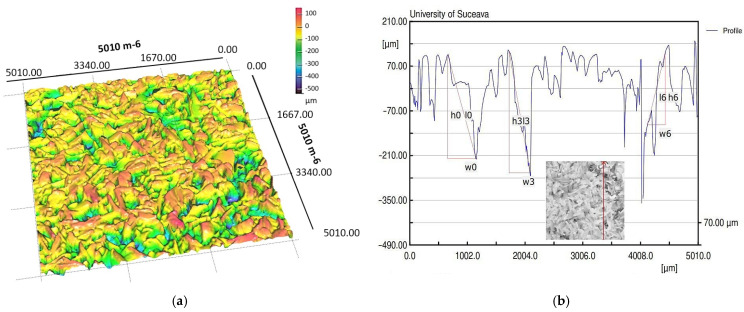
3D dry porous layer images (**a**) and sample with dimensions (**b**) obtained by laser profilometry.

**Figure 12 micromachines-14-01468-f012:**
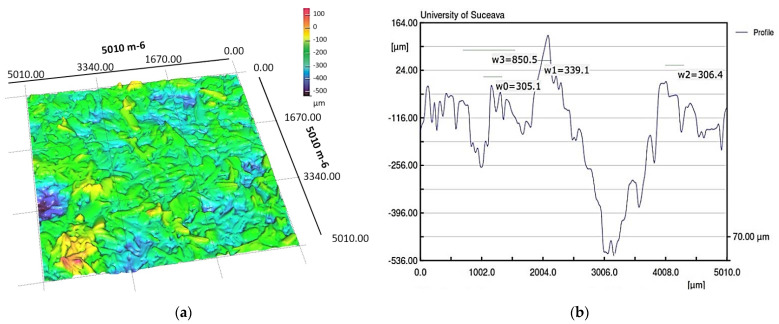
Images of the wetted porous layer in 3D (**a**) and the dimensions of the formed micro-channels (**b**), obtained by laser profilometry.

**Figure 13 micromachines-14-01468-f013:**
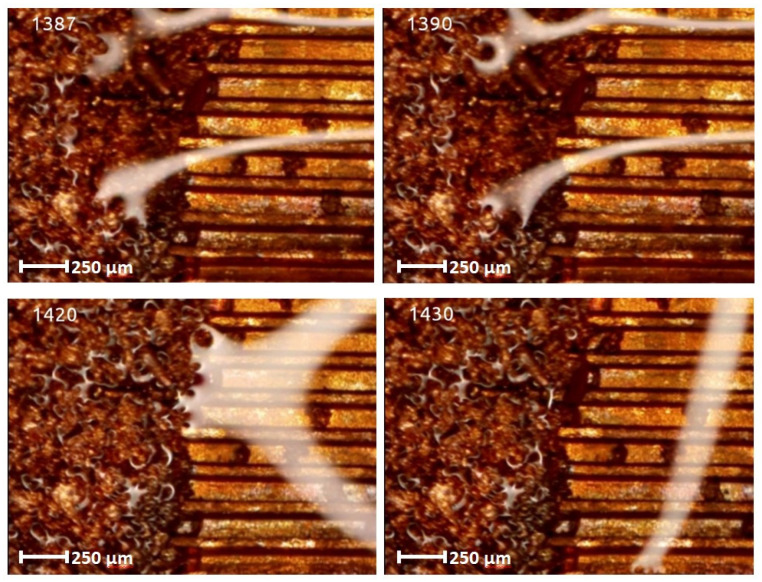
Vaporization zone with porous material and trapezoidal microchannels.

**Figure 14 micromachines-14-01468-f014:**
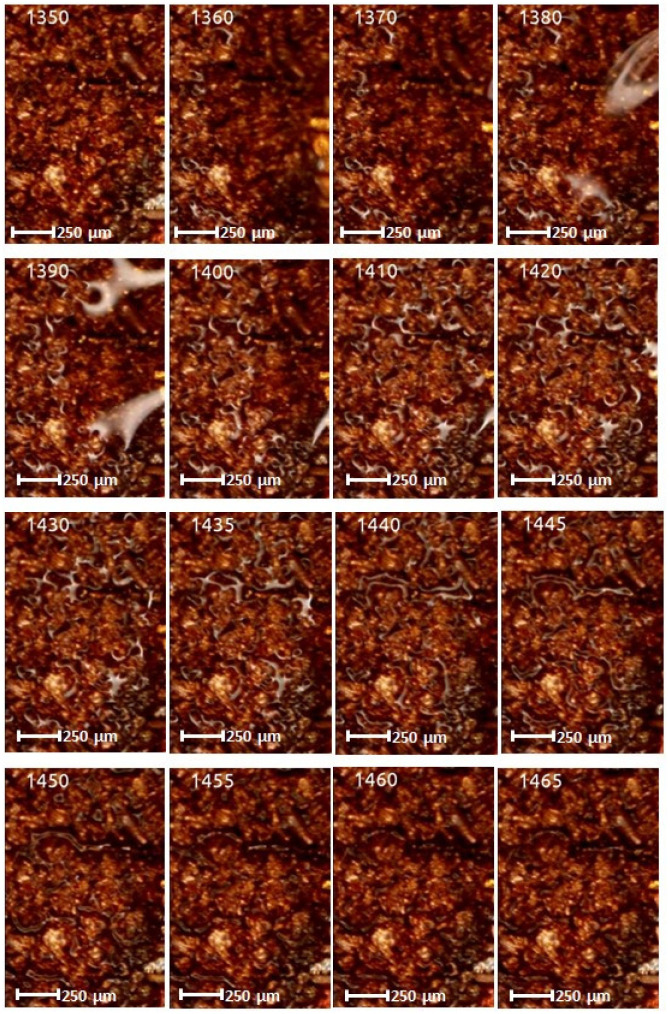
Frames with vapor formation and displacement in a porous medium in an FMHP.

**Figure 15 micromachines-14-01468-f015:**
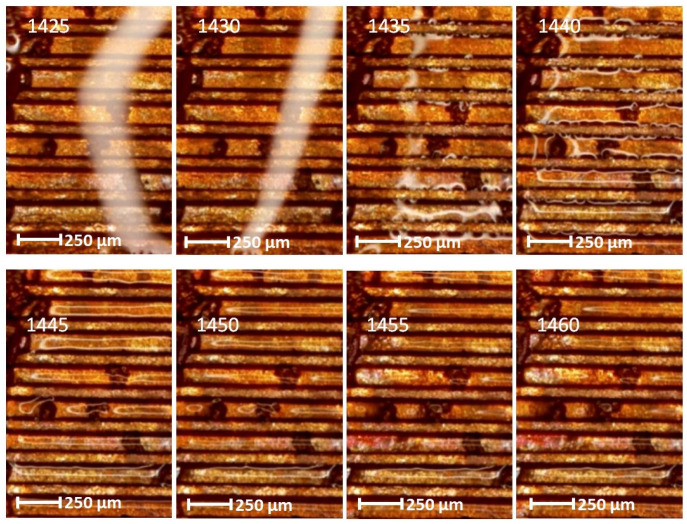
Frames with vapor formation and displacement in a porous medium in a micro heat pipe.

**Table 1 micromachines-14-01468-t001:** Polysynthetic layer microchannel profile values measured by laser profilometry.

	Measured Parameter	Length [mm]	Points –	Minimum [μm]	Maximum [μm]	Mean [μm]	Std-Dev. [μm]
Case							
Dry porous material		5.010	501	−401.945	148.977	13.208	88.085
Wetted porous material		5.010	501	−522.357	126.796	156.629	135.783

**Table 2 micromachines-14-01468-t002:** Considered values for capillary layers.

**Material**	**Pore Area** $\left[m^{2}\right]$	**Microsphere Diameter/Average Values of Copper Powder Particles** [m]	**Porosity** [%]
Capillary layer of sintered copper microspheres (average values)	$\left(5÷15\right)\times 10^{-8}$	$\left(52÷78\right)\times 10^{-6}$	35÷85
Porous sintered copper powder material (obtained by the authors)	$\left(98÷195\right)\times 10^{-9}$	$\left(10÷32\right)\times 10^{-6}$	25÷75

## Data Availability

Some or all data, models, or codes generated or used during the study are available from the corresponding author by request.

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
