# Peer review of "Assessment of Vapor Formation Rate and Phase Shift between Pressure Gradient and Liquid Velocity in Flat Mini Heat Pipes as a Function of Internal Structure"

_micromachines, 2023, doi:10.3390/mi14071468_

Round 1
Reviewer 1 Report

Minor editing of English language required
Author Response
"Please see the attachment."

Reviewer 2 Report
The review result of manuscript;
Micromachines: Manuscript micromachines-1250669
Title: Assessment of vapour formation rate and phase shift between pressure gradient and liquid velocity in flat mini heat pipes as a function of internal structure
Author : Ioan Mihai, Cornel Suciu, Claudiu Marian Picus
General comment;
This paper discusses the parameters affecting heat exchange capacity for FMHP using sintered porous copper material and trapezoidal microchannels. Parameters such as the microchannel hydraulic diameter and Reynolds number are given as a function of the porosity of the porous material, and the Nusselt number is obtained. The tube frictional coefficient of the porous media is calculated by various methods and analyzed as a function of porosity and Reynolds number, respectively. For the experiments, vapor formation rates for both porous media and trapezoidal microchannels have been determined and analyzed for heat transfer phenomena inside the FMHP.
Specific
The title of the paper does not adequately reflect the content of the report. The paper does not investigate thermal phenomena by parametrically changing the internal structure.
An illustration of the flow channel to be analyzed is required.
Similarly, a figure or illustration of the flow channel of the experimental apparatus is needed
The vibration velocity that appears in 3. needs to be explained.
Recommend checking documents by native English speakers. For instance, “In eq.(6), used to determine Cs” (line 133), “Both are interest in the study of” (line 442), etc.
Author Response
"Please see the attachment."

Reviewer 3 Report
Refer to the attached file

Acceptable English, but some of the sentences can be rewritten for better clarity.
Author Response
"Please see the attachment."

Round 2
Reviewer 1 Report
The revision provided by the Authors matches the recommendations indicated in the previous review. The Manuscript may be considered for publication in the current form.